# DeepUSPS: Deep Robust Unsupervised Saliency Prediction With Self-Supervision

**Duc Tam Nguyen** [\*†‡], **Maximilian Dax** [\*‡], **Chaithanya Kumar Mummadi** [†§]
**Thi Phuong Nhung Ngo** [§], **Thi Hoai Phuong Nguyen** [¶], **Zhongyu Lou** [‡], **Thomas Brox** [†]

## Abstract

Deep neural network (DNN) based salient object detection in images based on high-quality labels is expensive. Alternative unsupervised approaches rely on careful selection of multiple handcrafted saliency methods to generate noisy pseudo-ground-truth labels. In this work, we propose a two-stage mechanism for robust unsupervised object saliency prediction, where the first stage involves refinement of the noisy pseudo-labels generated from different handcrafted methods. Each handcrafted method is substituted by a deep network that learns to generate the pseudo-labels. These labels are refined incrementally in multiple iterations via our proposed self-supervision technique. In the second stage, the refined labels produced from multiple networks representing multiple saliency methods are used to train the actual saliency detection network. We show that this self-learning procedure outperforms all the existing unsupervised methods over different datasets. Results are even comparable to those of fully-supervised state-of-the-art approaches. The code is available at `https://tinyurl.com/wtlhgo3` .

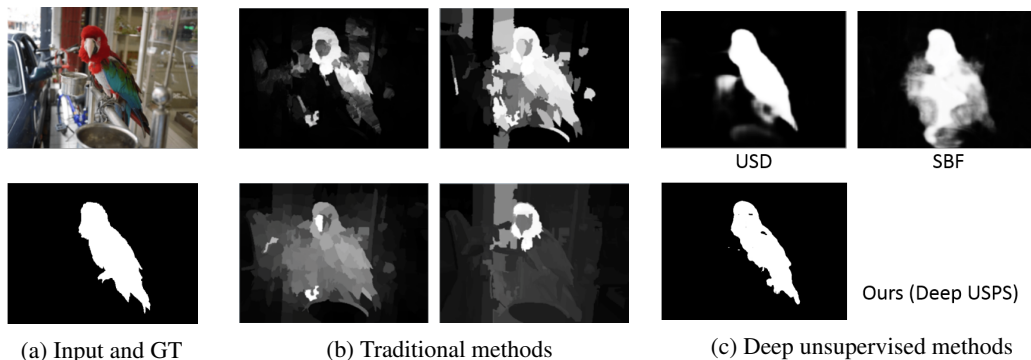

(a) Input and GT   (b) Traditional methods   (c) Deep unsupervised methods

Figure 1: Unsupervised object saliency detection based on a given (a) input image. Note that the ground-truth (GT) label is depicted only for illustration purposes and not exploited by any traditional or deep unsupervised methods. (b) Traditional methods use handcrafted priors to predict saliencies and (c) deep unsupervised methods SBF, USD and ours (DeepUSPS) employ the outputs of the handcrafted methods as pseudo-labels in the process of training saliency prediction network. It can be seen that while SBF results in noisy saliency predictions and USD produces smooth saliency maps, our method yields more fine-grained saliency predictions and closely resembles the ground-truth.

[\*]Equal contribution, [fixed-term.Maximilian.Dax, Ductam.Nguyen]@de.bosch.com

[†]Computer Vision Group, University of Freiburg, Germany

[‡]Bosch Research, Bosch GmbH, Germany

[§]Bosch Center for AI, Bosch GmbH, Germany

[¶]Karlsruhe Institute of Technology, Germany

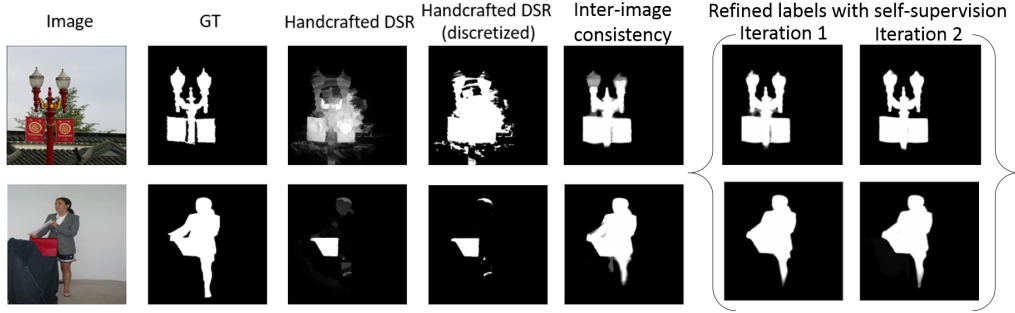

| Image | GT | Handcrafted DSR | Handcrafted DSR (discretized) | Inter-image consistency | Refined labels with self-supervision Iteration 1 | Iteration 2 |

Figure 2: Evolution of refined pseudo-labels from the handcrafted method DSR in our pipeline. Here, we show that the noisy pseudo label from the handcrafted method gets improved with inter-image consistency and further gets refined with our incremental self-supervision technique. While the perceptual differences between pseudo-labels from inter-image consistency and self-supervision technique are minor, we quantitatively show in Table 2 that this additional refinement improves our prediction results. Results from different handcrafted methods are depicted in Fig. 1 in Appendix.

# 1 Introduction

Object saliency prediction aims at finding and segmenting generic objects of interest and help leverage unlabeled information contained in a scene. It can contribute to binary background/foreground segmentation, image caption generation (Show, 2015), semantic segmentation (Long et al., 2015), or object removal in scene editing (Shetty et al., 2018). In semantic segmentation, for example, the network trained on a fixed set of class labels can only identify objects belonging to these classes, while object saliency detection can highlight an unknown object (e.g., "bear" crossing a street).

Existing techniques on the saliency prediction task primarily fall under supervised and unsupervised settings. The line of work of supervised approaches (Hou et al., 2017; Luo et al., 2017; Zhang et al., 2017b,c; Wang et al., 2017; Li et al., 2016; Wang et al., 2016; Zhao et al., 2015; Jiang et al., 2013b; Zhu et al., 2014) however, requires large-scale clean and pixel-level human-annotated datasets, which are expensive and time-consuming to acquire. Unsupervised saliency methods do not require any human annotations and can work in the wild on arbitrary datasets. These unsupervised methods are further categorized into traditional handcrafted salient object detectors (Jiang et al., 2013b; Zhu et al., 2014; Li et al., 2013; Jiang et al., 2013a; Zou & Komodakis, 2015) and DNN-based detectors (Zhang et al., 2018, 2017a). These traditional methods are based on specific priors, such as center priors (Goferman et al., 2011), global contrast prior (Cheng et al., 2014), and background connectivity assumption (Zhu et al., 2014). Despite their simplicity, these methods perform poorly due to the limited coverage of the hand-picked priors.

DNN-based approaches leverage the noisy pseudo-label outputs of multiple traditional handcrafted saliency models to provide a supervisory signal for training the saliency prediction network. Zhang et al. (2017a) proposes a method (SBF, 'Supervision by fusion') to fuse multiple saliency models to remove noise from the pseudo-ground-truth labels. This method updates the pseudo-labels with the predictions of the saliency detection network and yields very noisy saliency predictions, as shown in Fig. 1c. A slightly different approach (USD, 'Deep unsupervised saliency detection') is taken by Zhang et al. (2018) and introduces an explicit noise modeling module to capture the noise in pseudo-labels of different handcrafted methods. The joint optimization, along with the noise module, enables the learning of the saliency-prediction network to generate the pseudo-noise-free outputs. It does so by fitting different noise estimates on predicted saliency map, based on different noisy pseudo-ground-truth labels. This method produces smooth predictions of salient objects, as seen in Fig. 1c since it employs a noise modeling module to counteract the influence of noise in pseudo-ground-truth labels from handcrafted saliency models.

Both DNN-based methods SBF and USD performs direct pseudo labels fusion on the noisy outputs of handcrafted methods. This implies that the poor-quality pseudo-labels are directly used for training saliency network. Hence, the final performance of the network primarily depends upon the quality of chosen handcrafted methods. On the contrary, a better way is to refine the poor pseudo-labels in isolation in order to maximize the strength of each method. The final pseudo-labels fusion step

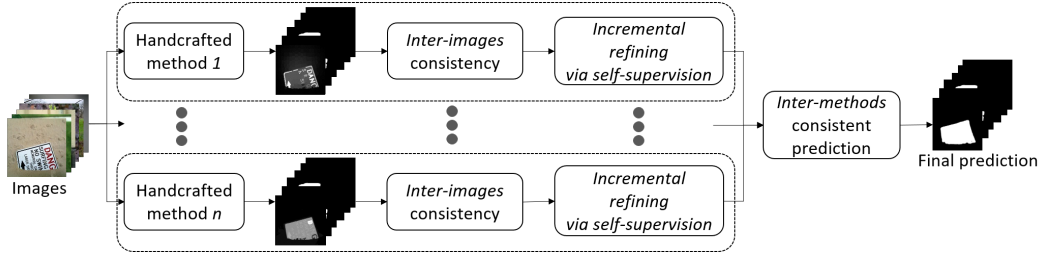

Figure 3: Overview of the sequence of steps involved in our pipeline. Firstly, the training images are processed through different handcrafted methods to generate coarse pseudo-labels. In the second step, which we refer to as *inter-images consistency*, a deep network is learned from the training images and coarse pseudo-labels to generate consistent label outputs, as shown in Fig. 2. In the next step, the label outputs are further refined with our self-supervision technique in an iterate manner. Lastly, the refined labels from different handcrafted methods are fused for training the saliency prediction network. Details of the individual components in the pipeline are depicted in Fig. 4.

to train a network should be performed on a set of diverse and high-quality, refined pseudo-labels instead.

More concretely, we propose a systematic curriculum to incrementally refine the pseudo-labels by substituting each handcrafted method with a deep neural network. The handcrafted methods operate on single image priors and do not infer high-level information such as object shapes and perspective projections. Instead, we learn a function or proxy for the handcrafted saliency method that maps the raw images to pseudo-labels. In other words, we train a deep network to generate the pseudo-labels which benefits from learning the representations across a broad set of training images and thus significantly improve the pseudo-ground-truth labels as seen in Fig. 2 (we refer this effect as inter-images consistency). We further refine our pseudo-labels obtained after the process of inter-image consistency to clear the remaining noise in the labels via the self-supervision technique in an iterative manner. Instead of using pseudo-labels from the handcrafted methods directly as Zhang et al. (2018, 2017a), we alleviate the weaknesses of each handcrafted method individually. By doing so, the *diversity* of pseudo-labels from different methods is preserved until the final step when all refined pseudo-labels are fused. The large diversity reduces the over-fitting of the network to the labels noise and results in better generalization capability.

The complete schematic overview of our approach is illustrated in Fig. 3. As seen in the figure, the training images are first processed by different handcrafted methods to create coarse pseudo-labels. In the second step, we train a deep network to predict the pseudo-labels (Fig. 4a) of the corresponding handcrafted method using a image-level loss to enforce inter-images consistency among the predictions. As seen in Fig. 2, this step already improves the pseudo-labels over handcrafted methods. In the next step, we employ an iterative self-supervision technique (Fig. 4c) that uses historical moving averages (MVA), which acts as an ensemble of various historical models during training (Fig. 4b) to refine the generated pseudo-labels further incrementally. The described pipeline is performed for each handcrafted method individually. In the final step, the saliency prediction network is trained to predict the refined pseudo-labels obtained from multiple saliency methods using a mean image-level loss.

Our contribution in this work is outlined as follows: we propose a novel systematic mechanism to refine the pseudo-ground-truth labels of handcrafted unsupervised saliency methods iteratively via self-supervision. Our experiments show that this improved supervisory signal enhances the training process of the saliency prediction network. We show that our approach improves the saliency prediction results, outperforms previous unsupervised methods, and is comparable to supervised methods on multiple datasets. Since we use the refined pseudo-labels, the training behavior of the saliency prediction network largely resembles supervised training. Hence, the network has a more stable training process compared to existing unsupervised learning approaches.

## 2 Related work

Various object saliency methods are summarized in Borji et al. (2014) and evaluated on different benchmarks (Borji et al., 2015). In the modern literature, the best performances are achieved by deep

supervised methods (Hou et al., 2017; Luo et al., 2017; Zhang et al., 2017b,c; Wang et al., 2017; Li et al., 2016; Wang et al., 2016; Zhao et al., 2015; Jiang et al., 2013b; Zhu et al., 2014) which all at least use some form of label information. The labels might be human-annotated saliency maps or the class of the object at hand. Compared to these fully- and weakly- supervised methods, our approach does not require any label for training. Our method can hence generalize to new datasets without having access to the labels.

From the literature of deep unsupervised saliency prediction, both Zhang et al. (2018, 2017a) use saliency predictions from handcrafted methods as pseudo-labels to train a deep network. Zhang et al. (2018) proposes a datapoint-dependent noise module to capture the noise among different saliency methods. This additional noise module induces smooth predictions in desired saliency maps. Croitoru et al. (2019) use an ensemble of teacher models to choose high-quality maps for the fusion steps. Zhang et al. (2017a) defines a manual fusion strategy to combine the pseudo-labels from handcrafted methods on super-pixel and image levels. The resulting, combined labels are a linear combination of existing pseudo-labels. This method updates the pseudo-labels with the predictions of a saliency detection network and yields very noisy saliency predictions. In contrast, we refine the pseudo-labels for each handcrafted method in isolation, and hence the diversity of the pseudo-labels is preserved until the last fusion step.

The idea of using handcrafted methods for a pseudo-labels generation has also been adapted by Makansi et al. (2018) for optical flow prediction. They introduce an assessment network to predict the pixel-wise error of each handcrafted method. Subsequently, they choose the pixel-wise maps to form the best-unsupervised saliency maps. These maps are used as data augmentation for a new domain. However, the best maps are bounded by the quality of existing noisy-maps from the handcrafted methods. In contrast to their work, our method improves individual methods gradually by enforcing inter-images consistency, instead of choosing pseudo-labels from the existing set. Further, their method fuses the original pseudo-labels directly in a single step. On the contrary, our fusion step is performed on the *refined* pseudo-labels in a late-stage to preserve diversity.

From the robust learning perspective, Nguyen et al. (2019b) proposes a robust way to learn from wrongly annotated datasets for classification tasks. These techniques can be combined with our presented method to improve the performance further. These advances also improve one-class-training use cases such as anomaly detection (Nguyen et al., 2019a), where the models are typically sensitive to noisy labeled data.

Compared to all previous unsupervised saliency methods, we are the first to improve the saliency maps from handcrafted methods in isolation successfully. Furthermore, our proposed incremental refining with self-supervision via historical model averaging is unique among this line of research.

## 3 DeepUSPS: Deep Unsupervised saliency prediction via self-supervision

In this section, we explain the technical details of components in the overall pipeline shown in Fig. 3.

### 3.1 Enforcing inter-images consistency with image-level loss

Handcrafted saliency predictions methods are consistent within an image due to the underlying image priors, but not necessarily consistent across images. They only operate on single image priors and do not infer high-level information such as object shapes and perspective projections. Such inter-images-consistency can be enforced by using outputs from each method as pseudo-labels for training a deep network with an image-level loss. Such a process leads to a refinement of the pseudo-labels suggested by each handcrafted method.

Let $D$ be the set of training examples and $M$ be a handcrafted method. By $M(x, p)$ we denote the output prediction of method $M$ over pixel $p$ of image $x \in D$. To binarize $M(x, p)$, we use simple function $l(x, p)$ with threshold $\gamma$ such that: $l(x, p) = 1$ if $M(x, p) > \gamma$; $l(x, p) = 0$, otherwise. $\gamma$ equals to $1.5 * \mu_{saliency}$ of the handcrafted method. This discretization scheme counteracts the method-dependent dynamics in predicting saliency with different degree of uncertainties. The discretization of pseudo-labels make the network less sensitive to over-fitting to the large label noise, compared to fitting to continuous, raw pseudo-labels.

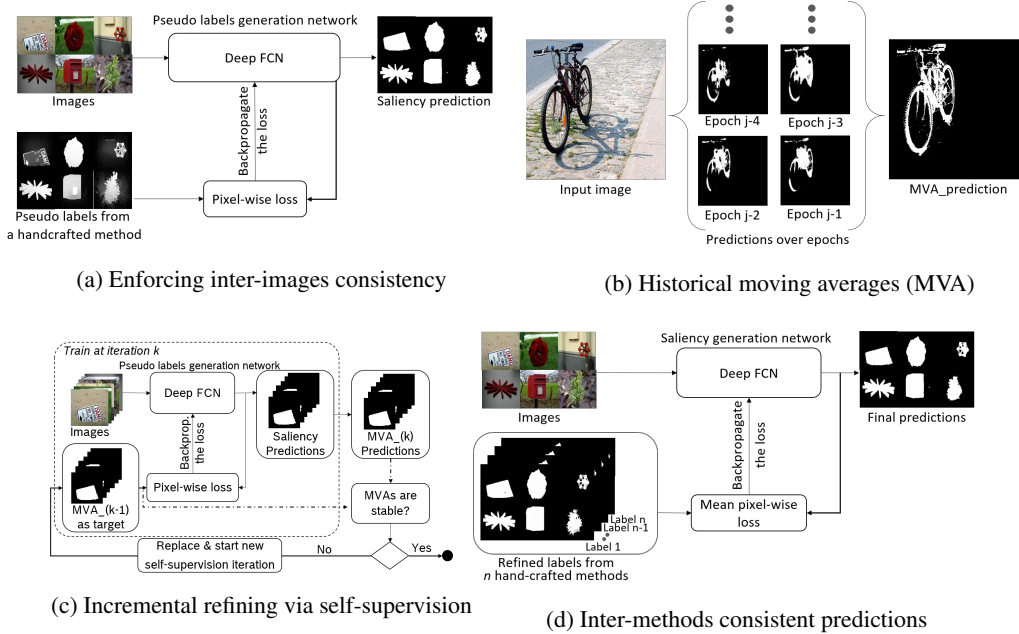

(a) Enforcing inter-images consistency

(b) Historical moving averages (MVA)

(c) Incremental refining via self-supervision

(d) Inter-methods consistent predictions

Figure 4: A detailed demonstration of each step in our pipeline from Fig. 3. Handcrafted methods only operate on single images and provide poor-quality pseudo-labels. Hence, (a)-(c) are performed for each handcrafted method separately to refine the pseudo-labels with deep network training. In the final stage (d), the refined pseudo-labels sets are fused by training a network to minimize the averaged loss between different methods.

Given method $M$, let $\theta$ be the set of its corresponding learning parameters in the corresponding FCN and $y(x, p)$ be the output of pixel $p$ in image $x$ respectively. The precision and recall of the prediction over image $x$ w.r.t. The pseudo-labels are straightforward and can be found in the Appendix. The *image-level loss* function w.r.t. each training example $x$ is then defined as $L_\beta = 1 - F_\beta$ where the F-measure $F_\beta$ reflects the weighted harmonic mean of precision and recall such that:

$$F_\beta = \left(1 + \beta^2\right) \frac{\text{precision} \cdot \text{recall}}{\beta^2 \, \text{precision} + \text{recall}}.$$

$L_\beta$ is a linear loss and therefore is more robust to outliers and noise compared to high-order losses such as Mean-Square-Error. The loss is to be minimized by training the FCN for a fixed number of epochs. The fixed number is small to prevent the network from strong over-fitting to the noisy labels.

**Historical moving averages of predictions**   Due to the large noise ration in the pseudo-labels set, the model snapshots in each training epoch fluctuates strongly. Therefore, a historical moving average of the network saliency predictions $y(x, p)$ is composed during the training procedure, as shown in Fig. 4b. Concretely, a fully-connected conditional random field (CRF) is applied to $y(x, p)$ after each forward pass during training. These CRF-outputs are then accumulated into MVA-predictions for each data point at each epoch $k$ as follows:

$$\text{MVA}(x, p, k) = (1 - \alpha) * CRF(y^j(x, p)) + \alpha * \text{MVA}(x, p, k - 1)$$

Since the MVA is collected during the training process after each forward pass, they do not require additional forward passes for the entire training set. Besides, the predictions are constructed using a large historical model ensemble, where all models snapshots of the training process contribute to the final results. Due to this historical ensembling of saliency predictions, the resulting maps are more robust and fluctuate less strongly compared to taking direct model snapshots.

## 3.2   Incremental pseudo-labels refining via self-supervision

The moving-average predictions have significantly higher quality than the predictions of the network due to (1) the use of large model ensembles during training and (2) the application of fully connected-CRF. However, the models from the past training iterations in the ensemble are weak due to strong fluctuations, which is a consequence of the training on the noisy pseudo-labels.

To improve the individual models in the ensemble, our approach utilizes the MVA again as the new set of pseudo-labels to train on (Fig. 4c). Concretely, the network is reinitialized and minimize the $L_\beta$ again w.r.t. MVA from the last training stage. The process is repeated until the MVA predictions have reached a stable state. By doing so, the diversity in the model ensemble is reduced, but the quality of each model is improved over time. We refer to this process as self-supervised network training with moving average (MVA) predictions.

### 3.3 Inter-methods consistent saliency predictions

Note, that the processes from Fig. 4a to Fig. 4c are applied to refine the outputs from each handcrafted method individually. These steps are intended to refine the quality of each method while retaining the underlying designed priors. Furthermore, refining each method in isolation increases the diversity among the pseudo-labels. Hence, the diversity of pseudo-labels is preserved until the final fusion stage. In the last step (Fig. 4d), the refined saliency maps are fused by minimizing the following loss:

$$L_{en} = \frac{1}{n}\Sigma_i L_\beta^i$$

where $L_\beta^i$ is computed similarly as aforementioned $L_\beta$ using the refined pseudo-labels of method $M_i$; and $\{M_1, \dots, M_n\}$ are the set of refined handcrafted methods. This fusion scheme is simple and can be exchanged with those from Zhang et al. (2018, 2017a); Makansi et al. (2018).

Our pipeline requires additional computation time to refine the handcrafted methods gradually. Since the training is done in isolation, the added complexity is linear in the number of handcrafted methods. However, the computation of MVAs does not require additional inference steps, since they are accumulated over the training iterations.

## 4 Experiments

We first compare our proposed pipeline to existing benchmarks by following the configuration of Zhang et al. (2018). Further, we show in detailed oracle and ablation studies how each component of the pipeline is crucial for the overall competitive performance. Moreover, we analyze the effect of the proposed self-supervision mechanism to improve the label quality over time.

### 4.1 Experiments setup

Our method is evaluated on traditional object saliency prediction benchmarks (Borji et al., 2015). Following Zhang et al. (2018), we extract handcrafted maps from MSRA-B (Liu et al., 2010): 2500 and 500 training and validation images respectively. The remaining test set contains in total 2000 images. Further tests are performed on the ECCSD-dataset (Yan et al., 2013) (1000 images), DUT (Yang et al., 2013) (5168 images), SED2 (Alpert et al., 2011)(100 images). We re-size all images to 432x432.

We evaluate the proposed pipeline against different supervised methods, traditional unsupervised methods and deep unsupervised methods from the literature. We follow the training configuration and setting of the previous unsupervised method Zhang et al. (2018) to train the saliency detection network. We use the DRN-network (Chen et al., 2018) which is pretrained on CityScapes (Cordts et al., 2016). The last fully-convolutional layer of the network is replaced to predict a binary saliency mask. Our ablation study also test ResNet101 (He et al., 2016) that is pretrained on ImageNET ILSVRC (Russakovsky et al., 2015). Our pseudo generation networks are trained for a fixed number of 25 epochs for each handcrafted method and saliency detection network is trained for 200 epochs in the final stage. We use ADAM (Kingma & Ba, 2014) with a momentum of 0.9, batch size 20, a learning rate of 1e-6 in the first step when trained on the handcrafted methods. The learning rate is doubled every time in later self-supervision iteration. Self-supervision is performed for two iterations. Our models are trained for three times to report the mean and standard deviation. Our proposed pipeline needs about 30 hours of computation time on four Geforce Titan X for training.

For the handcrafted methods, we use RBD ('robust background detection') (Zhu et al., 2014), DSR ('dense and sparse reconstruction') (Li et al., 2013), MC ('Markov chain') (Jiang et al., 2013a), HS ('hierarchy-associated rich features') (Zou & Komodakis, 2015). The $\alpha$-parameter for the exponential moving average for MVA maps is set to 0.7. Further, the model's predictions are fed into a fully-connected CRF (Krähenbühl & Koltun, 2011). As the evaluation metrics, we utilized

Mean-Average-Error (MAE or L1-loss) and weighted F-score with a $\beta^2 = 0.3$ similar to previous works. Furthermore, the analysis of the self-supervision mechanism includes precision and recall that compared against ground-truth-labels. Please refer to Sec. 1 in the Appendix for more details on the definition of these metrics.

## 4.2 Evaluation on different datasets

Tab. 1 shows the performance of our proposed approach on various traditional benchmarks. Our method outperforms other deep unsupervised works consistently on all datasets by a large margin regarding the MAE. Using the F-score metric, we outperform the state-of-the-art (noise modeling from Zhang et al. (2018) on three out of four datasets. Across the four datasets, our proposed baseline achieves up to 21% and 29% error reduction on the F-score and MAE-metric, respectively. The effects of different components are to be analyzed in the subsequent oracle test, ablation study, and detailed improvement analysis with self-supervision. Some failure cases are shown in Fig 5.

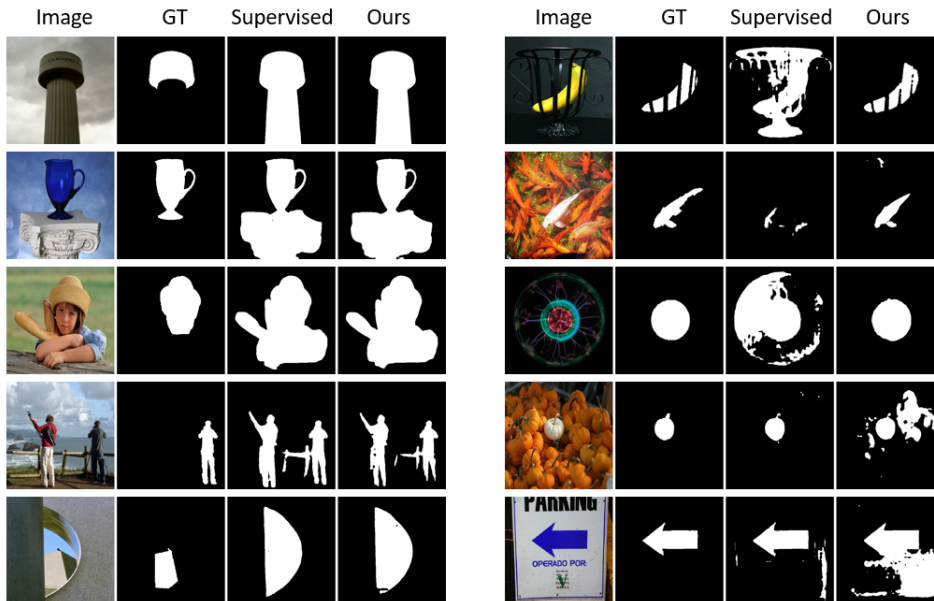

Figure 5: Failure Cases. The left panel shows images (first column) for which both, our approach (fourth column) and the supervised baseline (third column), fail to predict the GT label (second column). In each of these cases, both predictions are close to each other and visually look like justifiable saliency masks despite being significantly different than GT. We found that these kinds of images are indeed responsible for a major part of the bad scores. The right panel shows images for which our predictions are particularly good compared to the baseline prediction, or vice versa. These images are often disturbed by additional intricate details.

## 4.3 Oracle test and ablation studies

Tab. 2 shows an oracle test and an ablation study when a particular component of the proposed pipeline is removed. In the oracle test, we compare the training on ground-truth and oracle labels fusion in the final step, where we choose the pixel-wise best saliency predictions from the refined pseudo-labels. The performance of the oracle labels fusion is on-par with training on the ground-truth, or even slightly better on MSRA-B and SED2. This experiment indicates that DeepUSPS leads to high-quality pseudo-labels. Despite the simple fusion scheme, DeepUSPS approach is only slightly inferior to the oracle label fusion. Interchanging the architecture to ResNet101, which is pretrained on ImageNet ILSVRC, results in a similarly strong performance.

The ablation study shows the importance of the components in the pipeline, namely the inter images consistency training and the self-supervision-step. Training on the pseudo-labels from handcrafted methods directly causes consistently poor performance on all datasets. Gradually improving the particular handcrafted maps with our network already leads to substantial performance improvement.

Table 1: Comparing our results against various approaches measured in % of F-score (higher is better) and % of MAE (lower is better). Bold entries represent the best values in unsupervised methods.

| Models | MSRA-B | | ECSSD | | DUT | | SED2 | |
|---|---|---|---|---|---|---|---|---|
| | $F\uparrow$ | MAE$\downarrow$ | $F\uparrow$ | MAE$\downarrow$ | $F\uparrow$ | MAE$\downarrow$ | $F\uparrow$ | MAE$\downarrow$ |
| Deep and Supervised | | | | | | | | |
| Hou et al. (2017) | 89.41 | 04.74 | 87.96 | 06.99 | 72.90 | 07.60 | 82.36 | 10.14 |
| Luo et al. (2017) | 89.70 | 04.78 | 89.08 | 06.55 | 73.60 | 07.96 | - | - |
| Zhang et al. (2017b) | - | - | 88.25 | 06.07 | 69.32 | 09.76 | 87.45 | 06.29 |
| Zhang et al. (2017c) | - | - | 85.21 | 07.97 | 65.95 | 13.21 | 84.44 | 07.42 |
| Wang et al. (2017) | 85.06 | 06.65 | 82.60 | 09.22 | 67.22 | 08.46 | 74.47 | 11.64 |
| Li et al. (2016) | - | - | 75.89 | 16.01 | 60.45 | 07.58 | 77.78 | 10.74 |
| Wang et al. (2016) | - | - | 84.26 | 09.73 | 69.18 | 09.45 | 76.16 | 11.40 |
| Zhao et al. (2015) | 89.66 | 04.91 | 80.61 | 10.19 | 67.15 | 08.85 | 76.60 | 11.62 |
| Jiang et al. (2013b) | 77.80 | 10.40 | 80.97 | 10.81 | 67.68 | 09.16 | 76.58 | 11.71 |
| Zhu et al. (2014) | 89.73 | 04.67 | 83.15 | 09.06 | 69.02 | 09.71 | 78.40 | 10.14 |
| Unsupervised and handcrafted | | | | | | | | |
| RBD | 75.08 | 11.71 | 65.18 | 18.32 | 51.00 | 20.11 | 79.39 | 10.96 |
| DSR | 72.27 | 12.07 | 63.87 | 17.42 | 55.83 | 13.74 | 70.53 | 14.52 |
| MC | 71.65 | 14.41 | 61.14 | 20.37 | 52.89 | 18.63 | 66.19 | 18.48 |
| HS | 71.29 | 16.09 | 62.34 | 22.83 | 52.05 | 22.74 | 71.68 | 18.69 |
| Deep And Unsupervised | | | | | | | | |
| SBF | - | - | 78.70 | 08.50 | 58.30 | 13.50 | - | - |
| USD | 87.70 | 05.60 | **87.83** | 07.04 | 71.56 | 08.60 | 83.80 | 08.81 |
| DeepUSPS (ours) | **90.31** | **03.96** | 87.42 | **06.32** | **73.58** | **06.25** | **84.46** | **06.96** |
| $\pm$ | 00.10 | 00.03 | 00.46 | 00.10 | 00.87 | 00.02 | 01.00 | 00.06 |

The performance further increases with more iterations of self-supervised training. Leaving out the self-supervision stage also decreases the performance of the overall pipeline.

### 4.4 Analyzing the quality of the pseudo label

Fig. 6 shows an analysis of the quality of the labels of training images over different steps of our pipeline. We analyze the quality of the generated saliency maps (pseudo labels) from the deep networks and also the quality of aggregated MVA maps. Here, the quality of the pseudo labels is measured using the ground-truth labels information of the training set. It can be seen in the figure that the quality of the labels is improved incrementally at each step of our pipeline. Moreover, the quality of MVA maps is shown to be improved rapidly when compared with the saliency maps. Our self-supervision technique further aids in improving the quality of the labels slightly. After few iterations of self-supervision, the F-score and the MAE-score stagnate due to the stable moving-average predictions, and the saliency outputs maps also reach the quality level of the MVA-maps. Hence, in the case of offline-testing (when all test data are available at once), the entire proposed procedure might be used to extract high-quality saliency maps. In addition, the precision and recall of the quality of the labels are shown in Fig 2 in the Appendix. The handcrafted methods vary strongly in terms of precision as well as recall. This significant variance indicates a large diversity among these pseudo-labels. Our approach is capable of improving the quality of the pseudo labels of each method in isolation. Thus, the diversity of different methods is preserved until the last fusion step, which enforces inter-methods consistent saliency predictions by the deep network.

## 5 Conclusion

In this work, we propose to refine the pseudo-labels from different unsupervised handcraft saliency methods in isolation, to improve the supervisory signal for training the saliency detection network. We learn a pseudo-labels generation deep network as a proxy for each handcraft method, which further enables us to adapt the self-supervision technique to refine the pseudo-labels. We quantitatively show that refining the pseudo-labels iteratively enhances the results of the saliency prediction network and

Table 2: Results on extensive ablation studies analyzing the significance of different components in our pipeline using F-score and MAE on different datasets. Our study includes oracle training on GT, oracle label fusion - best pixel-wise choice among different pseudo label maps, using only the pseudo-labels of a single handcrafted method and also analyzing the influence of self-supervision technique over iterations.

| Models | MSRA-B | | ECSSD | | DUT | | SED2 | |
|---|---|---|---|---|---|---|---|---|
| | $F\uparrow$ | MAE$\downarrow$ | $F\uparrow$ | MAE$\downarrow$ | $F\uparrow$ | MAE$\downarrow$ | $F\uparrow$ | MAE$\downarrow$ |
| DeepUSPS (ours) | **90.31** | **03.96** | 87.42 | **06.32** | **73.58** | **06.25** | **84.46** | **06.96** |
| DeepUSPS (ours)-Resnet101 | 90.05 | 04.17 | 88.17 | 06.41 | 69.60 | 07.71 | 82.60 | 07.31 |
| (Oracle) train on GT | 91.00 | 03.37 | 90.32 | 04.54 | 74.17 | 05.46 | 80.57 | 07.19 |
| (Oracle) Labels fusion using GT | 91.34 | 03.63 | 88.80 | 05.90 | 74.22 | 05.88 | 82.16 | 07.10 |
| Direct fusion of handcrafted methods | 84.57 | 06.35 | 74.88 | 11.17 | 65.83 | 08.19 | 78.36 | 09.20 |
| *Effect of inter-images consistency training* | | | | | | | | |
| Trained on inter-images cons. RBD-maps | 84.49 | 06.25 | 80.62 | 08.82 | 63.86 | 09.17 | 72.05 | 10.33 |
| Trained on inter-images cons. DSR-maps | 85.01 | 06.37 | 80.93 | 09.28 | 64.57 | 08.24 | 65.88 | 10.71 |
| Trained on inter-images cons. MC-maps | 85.72 | 05.80 | 83.33 | 07.73 | 65.65 | 08.51 | 73.90 | 08.95 |
| Trained on inter-images cons. HS-maps | 85.98 | 05.58 | 84.02 | 07.51 | 66.83 | 07.83 | 71.45 | 08.43 |
| *Effect of self-supervision* | | | | | | | | |
| No self-supervision | 89.52 | 04.25 | 85.74 | 06.93 | 72.81 | 06.49 | 84.00 | 07.05 |
| Trained on refined RBD-maps after iter. 1 | 87.10 | 05.33 | 83.38 | 08.03 | 68.45 | 07.54 | 74.75 | 09.05 |
| Trained on refined RBD-maps after iter. 2 | 88.08 | 04.96 | 84.99 | 07.51 | 70.95 | 06.94 | 78.37 | 08.11 |
| Trained on refined DSR-maps after iter. 1 | 87.11 | 05.62 | 82.77 | 08.68 | 67.52 | 07.55 | 71.40 | 09.41 |
| Trained on refined DSR-maps after iter. 2 | 88.34 | 05.17 | 84.73 | 08.08 | 68.82 | 07.21 | 74.24 | 09.06 |
| Trained on refined MC-maps after iter. 1 | 87.53 | 05.22 | 84.94 | 07.58 | 67.82 | 07.33 | 70.72 | 09.48 |
| Trained on refined MC-maps after iter. 2 | 88.53 | 04.85 | 85.74 | 07.29 | 69.52 | 06.92 | 73.00 | 09.22 |
| Trained on refined HS-maps after iter. 1 | 88.23 | 04.73 | 86.21 | 06.66 | 71.21 | 06.63 | 76.75 | 07.80 |
| Trained on refined HS-maps after iter. 2 | 89.07 | 04.52 | 86.75 | 06.51 | 71.64 | 06.42 | 78.88 | 07.22 |

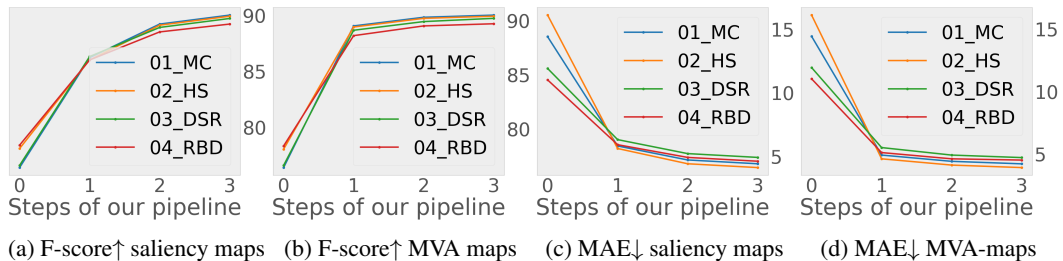

(a) F-score↑ saliency maps    (b) F-score↑ MVA maps    (c) MAE↓ saliency maps    (d) MAE↓ MVA-maps

Figure 6: Illustrating the improvement of labels quality of predicted saliency maps and aggregated MVA maps on MSRA-B training set from four handcrafted methods over different steps in our pipeline. The steps 0-3 represent measure on the quality of the labels of four different handcrafted methods, inter-images consistency, iteration 1 and iteration 2 of self-supervision with respect to the ground-truth labels.

outperforms previous unsupervised techniques by up to 21% and 29% relative error reduction on the F-score and Mean-Average-Error, respectively. We also show that our results are comparable to the fully-supervised state-of-the-art approaches, which explains that the refined labels are as good as human-annotations. Our studies also reveal that the proposed curriculum learning is crucial to improve the quality of pseudo-labels and hence achieve competitive performance on the object saliency detection tasks.

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
