[Supplementary Material · App_NeurIPS_19_Robust_Unsupervised_Saliency_Detection.pdf]

# DeepUSPS: Deep Robust Unsupervised Saliency Prediction With Self-Supervision –Supplementary Material–

**Duc Tam Nguyen** [*†‡], **Maximilian Dax** [*‡], **Chaithanya Kumar Mummadi** [†§]
**Thi Phuong Nhung Ngo** [§], **Thi Hoai Phuong Nguyen** [¶], **Zhongyu Lou** [‡], **Thomas Brox** [†]

## 1   Loss function

Due to the pseudo labels noise, the standard Cross-Entropy loss makes the model unstable, as it learns from outliers too strongly. Instead, we use the more robust image-level loss function

$$L_\beta = 1 - F_\beta, \tag{1}$$

where the F-measure $F_\beta$ is defined by the weighted harmonic mean of precision and recall according to

$$F_\beta = \left(1 + \beta^2\right) \frac{precision \cdot recall}{\beta^2 \, precision + recall}. \tag{2}$$

Precision quantifies how many of the predicted salient pixels are indeed pseudo-ground-truth salient, while recall specifies the fraction of the pseudo-ground-truth salient pixels that are also predicted to be salient. This translates to the following formulas,

$$precision = \frac{TP}{TP + FP} \qquad recall = \frac{TP}{TP + TN}, \tag{3}$$

where $TP$, $FP$ and $TN$ refer to True Positives, False Positives and True Negatives respectively. In case of discretized prediction $p$ and target $t$ they can be calculated as

$$
\begin{aligned}
TP &= \sum_i \left(p_i \cdot t_i\right) \\
FP &= \sum_i \left(p_i \cdot (1 - t_i)\right) \\
TN &= \sum_i \left((1 - p_i) \cdot t_i\right)
\end{aligned}
\tag{4}
$$

where the sum extends over all pixels $i$. A straightforward generalization to continuous predictions is achieved by dropping the constraint $p_i \in \{0, 1\} \forall i$ and allowing for continuous predictions $p \in [0, 1]$ instead. The targets remain discrete. This way, the F-measure and hence the loss is differentiable with respect to $p_i$ and can therefore be used for backpropagation.

---

[*]Equal contribution

[†]Computer Vision Group, University of Freiburg, Germany

[‡]Bosch Research, Bosch GmbH, Germany

[§]Bosch Center for AI, Bosch GmbH, Germany

[¶]Karlsruhe Institute of Technology, Germany

## 2 Samples of refined labels

Figure 1 shows several examples of pseudo labels refined in our pipeline.

(a) Input and label    (b) Discretized maps from trad. methods    (c) Refined saliency maps

Figure 1: Illustration of pseudo-labels that are generated in our pipeline after refining the coarse pseudo labels of four different traditional methods (MC, HS, RBD, and DSR presented in clockwise order starting from top-left). (a) shows the input image from the training set, (b) depicts the discretized pseudo labels of each handcrafted method and (c) shows the refined pseudo labels after two iterations of self-supervision in our pipeline.

# 3 More details of the ablation and oracle studies

Tab. 1 shows the ablation studies in more details.

Table 1: Results on extensive ablation studies analyzing the significance of different components in our pipeline for object saliency prediction. We measure the F-score (higher is better) and MAE (lower is better) on four different datasets. Here, oracle tests include the training on ground-truth (GT) labels and label fusion using GT - training on the best pixel-wise choice (measured using GT) among different pseudo label maps (this results in a single pseudo label map). We also analyzed the prediction results of the network that is trained only on pseudo labels of a single handcrafted method. Furthermore, we show the influence of self-supervision technique on the prediction results over iterations.

| Models | MSRA-B | | ECSSD | | DUT | | SED2 | |
|---|---|---|---|---|---|---|---|---|
| | $F\uparrow$ | MAE$\downarrow$ | $F\uparrow$ | MAE$\downarrow$ | $F\uparrow$ | MAE$\downarrow$ | $F\uparrow$ | MAE$\downarrow$ |
| DeepUSPS (ours) | **90.31** | **03.96** | 87.42 | **06.32** | **73.58** | **06.25** | **84.46** | **06.96** |
| ± | 00.10 | 00.03 | 00.46 | 00.10 | 00.87 | 00.02 | 01.00 | 00.06 |
| No CRF | 90.21 | 03.99 | 87.38 | 06.35 | 73.36 | 06.31 | 84.71 | 06.92 |
| ± | 00.12 | 00.03 | 00.13 | 00.04 | 00.21 | 00.08 | 00.45 | 00.08 |
| (Oracle) train on GT | 91.00 | 03.37 | 90.32 | 04.54 | 74.17 | 05.46 | 80.57 | 07.19 |
| ± | 00.10 | 00.03 | 00.46 | 00.10 | 00.87 | 00.02 | 01.00 | 00.06 |
| (Oracle) Labels fusion using GT | 91.34 | 03.63 | 88.80 | 05.90 | 74.22 | 05.88 | 82.16 | 07.10 |
| ± | 00.06 | 00.04 | 00.61 | 00.19 | 00.78 | 00.06 | 01.28 | 00.19 |
| Direct fusion of handcrafted methods | 84.57 | 06.35 | 74.88 | 11.17 | 65.83 | 08.19 | 78.36 | 09.20 |
| ± | 00.07 | 00.01 | 00.37 | 00.08 | 00.16 | 00.05 | 00.28 | 00.12 |
| Effect of inter-images consistency training | | | | | | | | |
| Trained on inter-images cons. RBD-maps | 84.49 | 06.25 | 80.62 | 08.82 | 63.86 | 09.17 | 72.05 | 10.33 |
| Trained on inter-images cons. DSR-maps | 85.01 | 06.37 | 80.93 | 09.28 | 64.57 | 08.24 | 65.88 | 10.71 |
| Trained on inter-images cons. MC-maps | 85.72 | 05.80 | 83.33 | 07.73 | 65.65 | 08.51 | 73.90 | 08.95 |
| Trained on inter-images cons. HS-maps | 85.98 | 05.58 | 84.02 | 07.51 | 66.83 | 07.83 | 71.45 | 08.43 |
| Effect of self-supervision | | | | | | | | |
| No self-supervision | 89.52 | 04.25 | 85.74 | 06.93 | 72.81 | 06.49 | 84.00 | 07.05 |
| Trained on refined RBD-maps after iter. 1 | 87.10 | 05.33 | 83.38 | 08.03 | 68.45 | 07.54 | 74.75 | 09.05 |
| Trained on refined RBD-maps after iter. 2 | 88.08 | 04.96 | 84.99 | 07.51 | 70.95 | 06.94 | 78.37 | 08.11 |
| Trained on refined DSR-maps after iter. 1 | 87.11 | 05.62 | 82.77 | 08.68 | 67.52 | 07.55 | 71.40 | 09.41 |
| Trained on refined DSR-maps after iter. 2 | 88.34 | 05.17 | 84.73 | 08.08 | 68.82 | 07.21 | 74.24 | 09.06 |
| Trained on refined MC-maps after iter. 1 | 87.53 | 05.22 | 84.94 | 07.58 | 67.82 | 07.33 | 70.72 | 09.48 |
| Trained on refined MC-maps after iter. 2 | 88.53 | 04.85 | 85.74 | 07.29 | 69.52 | 06.92 | 73.00 | 09.22 |
| Trained on refined HS-maps after iter. 1 | 88.23 | 04.73 | 86.21 | 06.66 | 71.21 | 06.63 | 76.75 | 07.80 |
| Trained on refined HS-maps after iter. 2 | 89.07 | 04.52 | 86.75 | 06.51 | 71.64 | 06.42 | 78.88 | 07.22 |

# 4 Self-supervision

Fig.. 2 shows how the quality of pseudo-labels for training samples are improved quantitatively. The quality is measured using precision and recall on the training set with respect to the *hold-out ground-truth labels*. Note that these ground-truth labels are not used for any training step. Here, the diversity among different methods can be seen clearly. Some methods are superior in terms of precision but inferior in terms of recall.

(a) Precision ↑ of saliency maps         (b) Recall ↑ of saliency maps

(c) Precision ↑ of MVA-maps         (d) Recall ↑ of MVA-maps

Figure 2: Illustrating the pseudo labels quality improvement by inter-images-consistency learning and self-supervision using the historical moving averages as new targets on the MSRA-B training set accessed using precision and recall. The scores are measured using the hold-out *ground-truth labels*, for network predicted pseudo labels (saliency output maps), and aggregated MVA maps (historical moving averages). Note that the ground truth labels are only used for measuring the quality of pseudo labels and not used during training. We stop the process of iterative self-supervision when the MVA-maps have stabilized, i.e., the changes in subsequent iterations are negligible. Here, the x-axis labels 0-3 represent measure on pseudo labels obtained at different stages in our pipeline in the following order: pseudo labels of handcrafted methods, inter-images-consistency training, refined pseudo labels from the first iteration of self-supervision and the pseudo labels from the second iteration of self-supervision.

# 5 Failure Cases

As shown in figure 3 the correlation between the performance of our unsupervised approach and the supervised (orcale) baseline is strong. In particular, there is a large overlap of failure cases).

Figure 3: Comparison of the MAE scores of our predictions (x-axis) and those of the baseline from the supervised setting (y-axis). The data indicates a strong correlation of the quality between the predictions of both settings.