[Reviews · NeurIPS 2019]

Reviewer 1



Saliency detection is an interesting and important problem in computer vision. It is often used for localising the object of interest in a given scene and as a guiding step (or an anchor) for object detection algorithms. Authors address this problem by adopting MAV/CRF and deep FCN in their framework. Although the outcome of this method is good, the authors do not motivate the use of FCN, CRF and handcrafted saliency prediction methods. Given that several state-of-the-art CNN based segmentation and saliency detection algorithms have been proposed in the recent literature, what value does this work add to CV/ML research? Can you please list the saliency detection methods whihc were used in this paper? Can you please comment on the computational efficiency of this technique compared to the other methods? A similar work, "Deep Unsupervised Saliency Detection: A Multiple Noisy Labeling Perspective" (CVPR 2018), with some overlap has been published in the past. How is your contribution different from this work?

Reviewer 2



Despite the impressive empirical results, the paper is difficult to follow, and it seems the proposed method is just a combination of existing previous methods. The paper, despite the simplicity of the method, could have used better writing. - in terms of style -- related work should come before describing the method. - when using acronyms (e.g. USD, SBF), for the first time, make sure the full name is mentioned as well, and the acronym in brackets. - if a method is referred to using an acronym (RBD, DSR, MC, HS) -- please use the same acronym in the tables -- it is difficult to trace the acronyms and citations throughout the paper otherwise. Are the results on ECSSD, DUT and SED2 evaluated with the model trained on MSRA-B? How would this method compare to Chen et al 2018 (DeepLab)? Is there a reason for not evaluating on Pascal segmentation? How does the weighted F-measure act as a pixel-wise loss, and how does it enforce inter-image consistency? The section (L86-106) is only explaining how to compute the F measure, and that loss is 1-F_beta. What is the difference between the "No-CRF" in the ablation in supplementary material and the "no self-supervision" in Table 2? L188-189 -- it is not clear what the change to the network was made. Is the same change applied to ResNet? If so, is only this new layer trained, and the rest of the network is frozen? Training seems to be done on a small number of images, for a small, fixed number of epochs (25). Is this sufficient to prevent overfitting? How was 25 decided? Did the authors consider other forms of regularization?

Reviewer 3



DeepUDPS is an interesting improvement to [Zhang et al. (2018, 2017a)]&[Makansi et al. (2018)] via at least 2 aspects: 1) refining noisy labels in isolation and then 2) incremental refining with self-supervision via historical model averaging and yields very competative results. Their method has generality to some extent e.g.,for other problem seetings including medical image segementation and so on, and from my viewpoint, though novelty is not specially significant due to some similarity to both the crowdsourcing and the unsupervised ensemble learning, the idea is intuitively simple yet effective. My other comments are 1.Using many handcrafted methods actually is similar to the crowdourcing in idea,where each handcrafted method plays a worker role for which the authors do mention nothing! 2. how to avoid cumulative mistakes once a mistake has truly been made but is not perceived. 3.One theoretical problem: Under what condition, such noisy label refinement can be helpful for better results? 4. Whether do the handcrafted methods need the diversity for improved performance? What effect is the number of the handcrafted methods used on performance? 5. For which images, your method will NOT work! 6.How do different initial handcrafted methods and the number of them influence on final label-quality, Do these methods need to selectively be fused? 7.Performance curves in Fig. 5 are NOT quite clear!

[Author Response · NeurIPS 2019]

**Reviewer #1** *Differences to Zhang et al., 2018 (USD)* : Important point! USD introduces an explicit noise modeling to
capture the noise and generate noise-free pseudo labels [L38-50]. This method produces blurry saliency predictions. On
the contrary, we refine the pseudo-labels individually to preserve diversity and enforce inter-image consistency before
fusing all the pseudo-labels, which is crucial for producing sharp and fine details of the salient objects.

*Computational efficiency*: The proposed framework needs extra computation for refining handcrafted methods in
isolation. However, the saliency prediction network converges faster than USD once the refined labels are available.

*List the saliency detection methods*: We use the handcrafted methods MC, HS, DSR, RBD [L197-198] in our work.

**Reviewer #1 & #2** *Value to CV/ML-research or technical contribution*: Existing unsupervised techniques combine
and reuse the handcrafted methods by directly adapting the noisy pseudo labels. We are the first to refine the labels
from these methods individually in isolation and incrementally improve them with self-supervision via historical model
averaging. This improves the results substantially, being on-par even with supervised learning while reducing the
manual labeling effort.

**Reviewer #2** *Modify style and acronyms*: To mitigate the confusion and improve the flow, we will place the related
works right after the introduction and incorporate the suggestions concerning the acronyms in our final version. Thanks
for pointing it out and helping us to improve the readability.

*Evaluation on ECSSD, DUT, SED2*: Yes, we follow the exact evaluation procedure of Zhang et al. 2018 and use the
model trained on MSRA-B to evaluate on these datasets. This is common practice for object saliency prediction.

*Compare to Chen et al. 2018 (DRN)*: DRN was developed for $n$-class semantic segmentation and was evaluated only on
this task. Inspired by its impressive results, we used DRN as backbone of our framework to make binary object saliency
predictions. We cannot compare our results on saliencies to semantic segmentation results from Chen et al.

*No evaluation on Pascal segmentation*: This dataset has non-binary labels (no binary ground-truth labels for object
saliency prediction) which impede the computation of the F-score measure.

*Is F-measure pixel-wise and how does it preserve inter-images consistency*: The F-measure is computed across the
pixels in the image and not pixel-wise. Thanks for pointing it out. We will remove the word "pixel-wise" from L102 to
avoid the confusion and reformulate this part. The pseudo-label generation network trained on entire dataset enforces
the inter-images consistency. The handcrafted methods do not leverage the features from images, whereas the deep
network learns to produce consistent output maps from the training images, as shown in Figure 2.

*'No-CRF' and 'no self-supervision' in tables*: No-CRF implies that we do not apply a CRF to the final outputs of the
network. This variant reduces the inference time for time-critical applications. No self-supervision indicates leaving out
"incremental refining via self-supervision" (Fig.4c) from the framework.

*Changes to DRN and ResNet*: The last layer of DRN produces multiple class outputs for semantic segmentation. We
modified this last layer to yield binary images, as needed for our saliency prediction framework, and trained the entire
network, including the last layer. Analogous changes are applied to ResNet.

*Why fix the no. of training iterations to 25*: We observed that network training reaches a coarse convergence on an error
plateau when combined with a small learning rate. Optimizing this hyper-parameter might lead to better performance.

*Other forms of regularization*: we investigated other techniques such as adversarial training, auxiliary losses with
inpainting or reconstruction. We found that minor improvement does not justify the added complexity of our system.

**Reviewer #3** *Mention connection to crowdsourcing*: Great suggestion! We will mention it in our final version.

*Avoid cumulative mistakes*: Given the labels' diversity among different handcrafted methods, the accumulated mistakes
are typically outnumbered in the final fusion step. It is unlikely that multiple methods make the same mistake.

*Under what condition noisy can label refinement be helpful for better results*: The noisy labels provide weak supervision,
which misleads the learning process and thus affects the network generalization. Refinement of the noisy labels improve
supervisory signal (similar to fully supervised setting), stabilizes training and enhances generalization of the network.

*Influence of the number of handcrafted methods on the final label quality:* The diversity of the pseudo-labels created
by different handcrafted methods is essential and actually more important than their absolute number. In Table 2, we
compare the performance of the full model to the saliency prediction network trained using labels attained from only a
single handcrafted method. The difference shows the importance of pseudo-labels from diverse methods.

*Failures cases:* We observe large overlapping with the traditional supervised learning methods in this regard. The
failures comprise corner cases like small objects and shadows. We will add failure cases in the final version.

*Do refined pseudo-labels need to be fused selectively?* Our framework shows that selective fusion is not necessary.
However, a clever fusion scheme may potentially further improve the system's performance.

*Fig. 5*: The curves (b & d) show the quality of MVA pseudo labels (the similarity of labels w.r.t. ground-truth) of every
handcrafted method at every step in our pipeline. The curves (a & c) show the differences in quality of saliency map
predictions obtained with the network trained on MVA pseudo labels retrieved at different steps in the pipeline.

[Meta-Review · NeurIPS 2019]

The paper presents a new approach DeepUSPS based on self-supervision and prediction consistency for saliency estimation. Results are very good and similar to supervised methods. At the beginning reviewers were no completely convinced on the goodness of the model which integrates several handcraft methods, but they agree on the fact that rebuttal was satisfying. Thus reviewers found consensus in the acceptance and the area chair agrees too-